# Experimental Ductility of Compression-Controlled Flexural Members Using CFRP Grid to Confine Concrete

**DOI:** 10.3390/ma14185163

**Published:** 2021-09-08

**Authors:** Antonis Michael, H. R. Hamilton

**Affiliations:** 1Department of Civil Engineering, School of Engineering, Frederick University, Nicosia 1036, Cyprus; 2Department of Civil and Coastal Engineering, University of Florida, Gainesville, FL 32611, USA; hrh@ce.ufl.edu

**Keywords:** composite materials, CFRP grid, concrete confinement, ductility, compression-controlled failure

## Abstract

Concrete members are typically designed so that flexural failure initiates with steel yielding and ends with concrete crushing in compression in order to take advantage of the yielding property of steel that allows for large deformations prior to any fracture of the material. On the other hand, if a large percentage of steel or linear elastic non-yielding reinforcement (i.e., FRP composite) is used, the member flexural failure typically initiates and ends with concrete crushing in compression. These members are known as compression-controlled members and typically exhibit brittle behavior. This study proposes a new approach in improving the flexural behavior of over-reinforced members through concrete confinement using carbon fiber reinforced polymer (CFRP) grid tubes in the compression zone. The concept was experimentally tested using rectangular beams. Beam 1 (control beam) had no grid reinforcement and beam 2 (tube beam) had two 152 mm grid tubes embedded in its compression zone. Experimental results indicate improvement in the ductility of the tube beam compared to the control beam of approximately 20–30% depending on the criteria used. Considering the low amount and mechanical properties of the CFRP grid, the improvement is significant, which shows that the proposed approach is valid and improves the ductility of compression-controlled members.

## 1. Introduction

Flexural failure of over-reinforced concrete members with steel tension reinforcement typically initiates and ends with concrete crushing in the compression zone. Therefore, such members lack the ductility that is associated with yielding of the reinforcement. This is typical not only for compression-controlled members with steel tension reinforcement but also for members utilizing linear-elastic non-yielding tension reinforcement, such as fiber-reinforced polymer (FRP) composite bars. Concrete as a material does not behave in a ductile manner. Concrete may exhibit limited ductility (also called pseudo-ductility), but this ductility varies depending on the aggregate material and the strength of the concrete used. High-strength concrete is typically more brittle than low-strength concrete and therefore, for high-strength concrete, the pseudo-ductility tends to diminish. 

International design codes [1,2] do not permit the design of flexural members that fail with a brittle manner because of the low ductility associated with such members. According to ACI 318 and EN 1992 [1,2], reinforced concrete members in bending can be classified either as balanced (steel yielding and concrete crushing take place at the same time), under-reinforced (steel yielding takes places prior to concrete crushing), and over-reinforced (concrete crushing takes places prior to steel yielding). ACI 318 ensures ductile response by requiring the steel strain in tension steel to reach at least 0.004 to ensure a tension-controlled failure. On the other hand, EN 1992, in order to achieve the same goal, limits the depth of the neutral axis χ to 0.45 d for concrete strengths of C50/60 or less, and 0.35 d for concrete strengths of C55/67 or higher.

In order to change the behavior of over-reinforced members, the properties of one of the main materials must be altered. The members must be designed in such a way so as to initiate and end failure in the material with the ability to undergo some plastic deformation. Since changing the properties of mild steel in an over-reinforced section will have no effect on the behavior (failure is taking place in concrete), the focus of improving member flexural behavior (ductility) should be placed on concrete rather than the tension reinforcement. One way to alter concrete material behavior is through confinement. If the concrete in the compression zone of a flexure dominated member is confined, then degradation of the compression zone at capacity is delayed. This results in a more ductile response.

Although the carbon fiber-reinforced polymer (CFRP) composite grid has been used as confinement reinforcement before, this was in the context of evaluating the concrete properties under axial compression in concrete standard cylindrical specimens [3,4] or in short piles in axial compression [5]. All researchers reported an increase in confined concrete strength and ultimate deformation at failure compared to the unconfined specimens. Seliem et al. [6] used a CFRP grid as an alternative to steel spirals for confining precast, prestressed concrete piles. They reported that one layer of the CFRP grid produced similar confinement as that provided by steel square spirals with a pitch of 75 mm. 

Carbon fiber-reinforced polymer composite grids have been used to reinforce concrete decks and beams for both strength and crack control [7,8,9,10,11], with significant improvements reported.

Several attempts have been made by various researchers to improve the ductility of over-reinforced concrete members. The success of the methods used varied. In some cases, helix reinforcement was used [12,13,14,15,16,17], in others, rectangular stirrups [16,17], and even a post-tensioned strap system was used to confine concrete [18]. Alasadi et al. used bolted steel plates of various thickness in the compression zone of over-reinforced concrete beams and recorded improvements in strength and ductility of the members [19,20]. All of these methods were used in order to avoid brittle failures associated with concrete crushing. Elbasha and Schmidt [12] investigated the effect on the ductility of high-strength reinforced concrete beams with helical reinforcement located in the compressive region of beams. The beams were loaded in flexure, and based on the results, the authors concluded that the helical reinforcement was an effective way of increasing ductility in high-strength reinforced concrete beams. They also reported that when the compressive strength of concrete increased, the failure yields and deflections were reduced. This was followed by a significant decrease in the ductility index. Others reported that an increase in the ratio of longitudinal reinforcement increased the displacement ductility index. This allowed higher ultimate displacements to take place in a beam [15]. 

Other findings on the installation of short stirrups as confinement reinforcement were also reported [21]. For example, Jeffery and Hadi reported that helical reinforcement is more effective compared to short stirrups in improving flexural capacity and the displacement ductility index of members utilizing confined concrete [13]. In other studies (Qin et al. [22]), hybrid systems (normal steel reinforcement and FRP bars) were investigated. Qin et al. reported that the hybrid reinforcement ratio between FRP and steel, A_f_/A_s_, had a significant influence on the flexural performance of hybrid FRPRC beams. The effect of the hybrid reinforcement ratio on the flexural performance of concrete beams was studied for both under- and over-reinforced scenarios using three-dimensional finite element models. The results showed that for a preferable strength and ductility performance to be obtained, an appropriate design of the hybrid reinforcement ratio is necessary for optimizing strength and ductility performance. It is obvious that the main objective in all of these studies with the implementation of a number of different techniques is the improvement of flexural capacity and stiffness. The aim is to prevent brittle failures in over-reinforced flexural members.

The application of a carbon FRP composite grid as confining reinforcement for concrete is investigated in this study. In order to verify the proposed concept, experimental testing was conducted on beams using a commercially available CFRP grid. The grid was formed into tubes and was embedded in concrete as confinement reinforcement in the compression zone. The beams were tested in a simply supported four-point bending configuration. The aim of the study was to change the behavior of the over-reinforced members under flexure and subsequently improve member ductility. The results indicate an improvement in strength, ductility (ductility index), and energy absorption.

Applications of the CFRP grid in over-reinforced members under flexure to improve member flexural behavior and produce a more ductile response are not available in the literature, and it appears they have not been attempted yet. Therefore, the data from this study will serve as the starting point for further studies in this research area. 

## 2. Materials and Material Testing

### 2.1. Carbon Fiber-Reinforced Polymer Grid

The CFRP composite grid is fabricated from carbon fibers embedded in an epoxy matrix. It is available in the form of rolls of 1.041 m wide and 274.32 m long (see Figure 1). The strand spacing in the longitudinal direction is approximately 46 mm, and in the transverse direction, 41 mm. The CFRP grid had an openness of 69%, which means that only 31% of the surface area is covered by the carbon fibers. This allows the embedment of the grid in concrete members.

### 2.2. Carbon Fiber-Reinforced Polymer Grid Testing

The tensile properties of the grid were obtained through testing of strands from both the longitudinal and transverse directions [3]. The specimen testing configuration is shown in Figure 2a. The strands were embedded into a steel pipe filled with an expansive grout. Strain was measured using strain gauges. The specimens were attached to the loading apparatus, as shown in Figure 2b. The loading apparatus consisted of a hydraulic actuator mounted to a stiff steel frame (Figure 2a) [3].

Table 1 shows the results of the tensile tests on the grid strands. Strands taken from the longitudinal direction were designated as longitudinal (L) and strands taken from the transverse direction were designated as transverse (T). The average values for strength and modulus were used as the properties of the grid.

The strength of each specimen was taken as the ratio of the peak load over the cross-sectional area and the modulus was determined by linear regression of the stress–strain data (Figure 3).

Although the number of tests may be considered as insufficient to determinate any statistical significance, the relatively high coefficient of variances (COV) for strength (24%) and tensile modulus (18%) indicate significant variability between the CFRP strands. The average grid strand strength was 696 MPa and the average modulus was 64 GPa. These values were significantly lower than the typical average values found in the literature for carbon epoxy unidirectional composites. The properties of the strands of this particular grid were approximately 2.5 times lower than the typical properties in the literature.

### 2.3. CFRP Grid Concrete Confinement Using Cylindrical Specimens

To evaluate the level of confinement provided by the CFRP grid, a test program involving the axial compressive testing of cylindrical specimens was conducted. Cylindrical specimens with and without the CFRP grid were tested [3]. The specimens were divided into control specimens (no grid reinforcement) and grid specimens (2 layers of CFRP grid reinforcement formed into a tubular (coil) configuration). The CFRP grid tubes were placed inside plastic cylinder molds (Figure 4). A limestone aggregate concrete mixture was used with a specified minimum compressive strength at 28 days of 31 MPa. All concrete cylindrical specimens were air-cured inside the plastic mold. The cylindrical specimens were capped using sulfur cement prior to testing. The specimens were tested using an MTS loading frame. Two linear variable displacement transducers (LVDTs) (see Figure 5) were used to record the length change of the specimens during loading [3].

Experimental results from the cylindrical specimens are listed in Table 2. The average strength for the control specimens tested in displacement control mode was 47.8 MPa and the COV was 2%. The average strength of the grid specimens was 52.9 MPa with a COV of 8.6%.

The control specimens failed in a typical manner without very significant post-peak behavior. Concrete crushing for the control specimens took place at a strain of approximately 0.0031. A control cylinder after testing can be seen in Figure 6a. Grid cylinders failed when CFRP grid strands ruptured (Figure 6b). Typically, the maximum strain at failure for the grid cylinders reached at least 0.005 for grid 1–3 specimens and 0.006 for grid 4–6 specimens.

Grid specimens reached higher peak loads and accommodated larger displacements compared to control specimens. This is evident in Figure 7, where typical experimental stress–axial strain curves for both control and grid specimens are plotted. The area under the stress–strain curve of the grid specimens was approximately 2.3 times larger compared to the area of the control specimens. The maximum average strain for the control specimens was 0.0031, while for the grid specimens it was 0.0053, which was approximately 1.7 times higher. All grid specimens reached a peak axial load followed by a descending post-peak curve. Other researchers (Harries and Kharel [23]) observed similar behavior in lightly confined concrete. One- and two-ply E-Glass confined cylinders had between a 10% and 20% increase in concrete strength [23], similar to the 11% increase observed in this study.

### 2.4. Concrete Confinement Model

When the CFRP grid is used as confinement reinforcement, the confining pressure and confinement effectiveness are less than that of a fully wrapped system. Therefore, only light confinement is achieved. Models developed using data from highly confined concrete may not be adequate. Several existing models were investigated to model the behavior of CFRP grid confined concrete. The grid strands in the circumferential direction only cover part of the area. One approach is to determine an equivalent full-coverage thickness for the circumferential grid strands. The equivalent grid thickness (*t_eg_*) was calculated based on the following expression:(1)$$t_{eg}=\frac{n_{l}\cdot b_{g}\cdot t_{g}}{s_{g}}$$
where *n_l_* is the number of CFRP grid layers, *b_g_* is the width of the grid strands, *t_g_* is the thickness of the grid strands, and *s_g_* is the CFRP grid strand spacing.

The secant modulus of elasticity of concrete (*E_c_*) was calculated based on the recommended expression used in Eurocode 2 [2]:(2)$$E_{c}=E_{cm}=22\;\cdot {\left[\frac{f_{cm}}{10}\right]}^{0.3}$$
where $f_{c}^{\prime }$ is the minimum specified compressive strength of concrete, in N/mm^2^.

The confinement strength (*f_ru_*) was determined using simple pressure vessel mechanics. The equilibrium condition requires that the force from the confining strength be equal to the force in the FRP encasement. The force from confinement is equal to the confining strength times the diameter of the enclosed concrete, and the force in the encasement is equal to the strength of the encasement times twice the thickness of the FRP encasement. By rearranging the equation, the confinement strength (*f_ru_*) was found:(3)$$f_{ru}=\frac{2\cdot t_{eg}}{d_{g}}\cdot f_{gu}$$
where *f_ru_* is the confinement strength, *d_g_* is the diameter of the CFRP grid tube, and *f_gu_* is the ultimate strength of the CFRP grid strands.

Assuming that the confined concrete is in a triaxial stress state, the increase in strength provided by the confinement is reflected in the maximum stress ($f_{cc}^{\prime \prime }$) for a cylindrical specimen, which is defined as [24]:(4)$$f_{cc}^{\prime \prime }=f_{c}^{\prime }+k_{1}\cdot f_{ru}$$
where *k*_1_ is the confinement effectiveness coefficient.

The confinement effectiveness coefficient for concrete confined by steel is usually taken between 2.8 and 4.1. Campione and Miraglia [25] found that the above values overestimate the confinement effectiveness coefficient for concrete wrapped with FRP. They found the confinement effectiveness coefficient for FRP-wrapped concrete to be 2. For the purpose of this study, the confinement effectiveness coefficient was taken as 2.

The axial strain of unconfined (Equation (5)) and CFRP grid confined (Equation (6)) concrete at the peak stress (*ε_co_*) was determined based on available expressions in the literature [4,26]:(5)$$\varepsilon_{co}=1.8\cdot \frac{f_{cc}^{\prime \prime }}{E_{c}}$$
(6)$$\varepsilon_{co}=\varepsilon_{cp}=\varepsilon_{co}^{\prime }\left(1+0.005\frac{f_{le}}{\varepsilon_{g}\sqrt{f_{co}^{\prime }}}\right)$$

The modified Hognestad stress–strain equations for the pre-peak (Equation (7)) and post-peak (Equation (8)) regions are as follow:(7)$$f_{c}=f_{cc}^{\prime \prime }\cdot \left[\frac{2\cdot \varepsilon_{c}}{\varepsilon_{co}}-{\left(\frac{\varepsilon_{c}}{\varepsilon_{co}}\right)}^{2}\right]$$
(8)$$f_{c}=f_{cc}^{\prime \prime }\cdot \left[1-D_{c}\cdot \left(\varepsilon_{c}-\varepsilon_{co}\right)\right]$$
where *ε_c_* is the concrete strain at a specific stress.

The material properties of the CFRP grid strands were used to develop the stress–strain curve of the CFRP grid confined concrete. The average strength of the control cylinders was taken as the strength of unconfined concrete ($f_{c}^{\prime }$). An average CFRP grid tube diameter of 136.5 mm was used. The deterioration constant was taken as 250 for unconfined concrete and 105 for confined concrete based on the experimental data. The experimental and theoretical stress–strain curves of CFRP grid confined concrete can be found in Figure 8.

The modified Hognestad curves matched well with the experimental curves. The average experimental peak stress for the unconfined concrete was 47.2 MPa, while the model obtained a peak stress of 47.8 MPa. In the case of grid confined concrete, the average experimental peak stress was 52.9 MPa, while the model predicted 52.8 MPa. The predicted values were almost identical compared to the experimental values, with a difference of less than 1.3%. The average experimental strain at the peak stress was 0.00239 for the unconfined concrete and 0.00259 for the confined concrete, with the model predictions at 0.00235 and 0.00264, respectively. The differences were again small, with the values almost identical (differences of 2% and 2.5%).

## 3. Compression-Controlled Concrete Beams with CFRP Grid Tubes in the Compression Zone

The unique application of the CFRP composite grid as confining reinforcement in the compression zone of compression-controlled beams was investigated with the construction and testing of two full-size beams. Beam 1 (control beam) had no CFRP composite grid tubes and beam 2 (tube beam) had two CFRP composite grid tubes placed in the compression zone. Details of the dimensions, amount, and location of steel reinforcement and CFRP composite grid tubes can be found in Figure 9 and Figure 10.

### 3.1. Materials and Beam Manufacturing

The mechanical properties of the CFRP composite grid were discussed in detail in previous sections. The CFRP composite grid was formed into a circular tube (or coil) using a 152 mm diameter plastic pipe, and was held in place by a thin string wrapped around the tube along its length (See Figure 11a).

The concrete used in both beams was ready-mix concrete with limestone aggregates. It had a 28-day design strength of 35 MPa, and maximum aggregate size of 9.5 mm. Mild steel reinforcement with minimum yield strength of 420 MPa was used for both tension and shear reinforcement.

Both beams were cast on the same day. Concrete was placed in both beams continuously and was vibrated as seen in Figure 11b. Concrete was sampled with the casting of 8 standard (150 × 300 mm) cylinders. The cylinders were placed in the laboratory and cured under the same conditions as the beams (air-dried curing) until testing day.

### 3.2. Test Set-Up and Instrumentation

Both beams were tested in a simply supported four-point bending configuration, as shown in Figure 12. The span length between the two supports was approximately 4.3 m. A hydraulic actuator was used to apply the load. The load was distributed to the two loading points using a steel spreader beam. From the spreader beam, the load was transferred to the beam using two steel rollers. The rollers allowed the loading mechanism to follow the rotation of the beam as the load increased. Steel plates distributed the load to avoid localized failure of the concrete material should the load be applied to a very small area. A load cell was used to measure the total applied force. The load points were located 1.7 m from each support. This created a 0.91 m region in the center of the span with a constant bending moment and zero shear force. This allowed the evaluation of beam behavior in pure flexure.

The beams were loaded at a constant rate of approximately 0.5 kN/s. The rate was selected in order to reach concrete crushing strain in approximately the same the time it took when the concrete cylinders were tested. The load was applied monotonically until the beam could no longer sustain a significant load, which for the purposes of this study was set to approximately 60% of the peak load.

A series of 15 displacement transducers (LVDTs) were placed at 305 mm intervals to measure the displacement of the beams at different locations along their length (see Figure 12). The LVDTs were labeled from north to south, as D1 through D15.

Foil strain gages were attached on the sides of the beams at mid-span (Figure 13a) and on top of the beam (Figure 13b). The strain gages at the side of the beams were labeled from the top down as SE1 through SE7 for the east side and SW1 through SW7 for the west side of the beam. The strain gages on the top of the beams were labeled from north to south as ST1 through ST3. The strain gages on the side of the beams were used to measure strain across the depth of the section, whereas the strain gages on top to measure the transverse expansion of the compression zone expected due to the Poisson effect. Data from all instrumentation devices were recorded using a data acquisition system at a rate of 4 Hz.

## 4. Experimental Results and Discussion

The concrete cylinders were tested on the same day as the beams in order to have an accurate value of the unconfined concrete strength in the beams. The results are listed in Table 3. The average compression strength for the cylinders tested was 41.8 MPa with a coefficient of variance of 2.3%.

Both beams failed due to concrete crushing in the constant moment region, as it can been seen in Figure 14. The compression zone after testing for the control beam can be seen in Figure 14a, while the one for the tube beam is shown in Figure 14b. It is evident from Figure 14 that the amount of damage to the compression zone of the control beam was more severe compared to that of the tube beam. The concrete inside the CFRP composite grid tubes maintained its shape inside the tube to a great extent, even after the tube ruptured and confinement was lost. The CFRP composite grid ruptured at specific locations and not throughout the constant moment region.

The load-displacement curves for both beams can be seen in Figure 15. The peak loads were approximately 658.3 kN for the control beam and 680.6 kN for the tube beam. There was a nominal increase in the peak load of the tube beam of approximately 3% compared to the control beam, which was considered insignificant. However, what was significant was the fact that the tube beam showed a relatively significant change in its behavior. The tube beam was able to undergo additional displacement beyond the peak load, something that was not the case with the control beam. The control beam only deflected an additional 1 mm when concrete began to crush at the extreme top fiber, while the tube beam deflected an additional 7 mm after reaching the peak load.

As it can be seen from Figure 15, the control beam loses the ability to support load almost immediately after reaching the peak load. On the other hand, for the tube beam, the load reduces by approximately 12% immediately after peak load, but the beam is able to support the rest of the load, as it can been seen by the small plateau that was created. The tube beam was able to undergo some plastic deformation before the CFRP grid started rupturing, which led to the crushing of concrete inside the CFRP grid tubes. The 12% load reduction immediately after the peak is due to concrete crushing of the top concrete (top 25 mm) in the compression zone which was not confined.

The strains measured from the strain gages installed on the two beam sides (east and west) were approximately the same for each specific location. This was typical for all strain gages. The strain at each location was calculated by taking the average value recorded by the two strain gages corresponding to each location. For example, the strain at 25 mm from the top of the beam (S1) was taken as the average of SE1 and SW1. The same approach was used for all other locations. The maximum average strain at peak load recorded during the test for the control beam was at the S1 location and was equal to 0.00284. The maximum average strain recorded at peak load for the tube beam was also at location S1 and was equal to 0.00255, approximately 10% lower than the strain recorded for the control beam. However, these were not the maximum values since it is known that maximum compressive strains occur at the extreme fibers, in this case located at the top of the beam. The compressive strain at the extreme concrete fiber can be redetermined by linear interpolation of the available experimental strain data. It should be noted that cracking at the location of the bottom strain gages prevented the recording of valid data by those gages, and that is the reason they were excluded from the plotting of the strain profiles. Assuming a linear distribution of strain through the beam cross-section, a straight line can be fitted, and the equation of the fitted line can be used to determine the strain at the extreme concrete fiber. The fitted lines were in very good agreement with the experimental data, as seen in Figure 16a. This was typical both for the control and tube beam. In order to estimate the crushing strain for the confined concrete in the CFRP grid, strain data from gages S4 and S5 were used for the load that caused concrete crushing in the CFRP tube. Data from gages S1–S3 were not included since they either stopped recording or did not functioning properly due to unconfined concrete crushing at their locations. The strain profile is shown in Figure 16b.

The strain at the extreme compression fiber (top) at peak load for the control beam was 0.00317, and 0.00301 for the tube beam. This was expected since limestone aggregate concrete exhibits higher brittleness compared to concretes with other types of aggregates. The ACI 318 concrete design code assumes a concrete crushing strain of 0.003 [1], which is in good agreement with the crushing strains calculated from the experimental data. Eurocode 2 on the other hand recommends a concrete crushing strain of 0.0035 for concretes with strength less than or equal to 50 MPa, which in this case is an overestimation [2]. The concrete strain at the top of the grid tube (location −25 mm) was 0.00499, as can be seen in Figure 16b. This compares well with the crushing strain of the grid cylindrical specimens, as discussed in Section 2.3, which was 0.0053. The difference between the two values was approximately 6%.

## 5. Experimental Moment Curvature Analysis of Tested Beams

The moment curvature (M-Φ) diagrams of the beams were developed using the available experimental data. It is well-known that the curvature can be determined by taking the second derivative of the deflection equation, as follows [27]:(9)$$\Phi =\frac{d^{2}y}{{\mathrm{dx}}^{2}}$$

Using the displacement values from the LVDTs at various loads and fitting a polynomial line through the deflection profile across the length of the beam, the deflection equation was developed at various increasing loads. The polynomial used was of the third degree since it is well-known that the displacement equations for point loads are third degree polynomial equations with respect to distance. An example is shown in Figure 17. The fitted lines were in good agreement with the displacement data (average R^2^ = 0.997). By taking the second derivative of the deflection equations, the curvature equations for each load were developed and used to calculate the curvature values. The moment at midspan was calculated for each load and was plotted against the curvature from zero load up to a post-peak load equal to 80% of the peak load. The curves for the mid-span location are shown in Figure 18. These curves show very similar trends as the load-displacement curves discussed earlier.

## 6. Ductility and Energy Absorption Capacity

The ductility, in the form of ductility factors, was calculated to investigate the structural behavior of the tested beams. The ductility of the specimens was calculated using two ductility factors, the deformation and curvature ductility factors. The deformation ductility factor was defined as the ratio of the axial deformation at 20% drop of axial load and axial deformation at yield load [28].

For the purpose of this study, the yield load definition suggested by Pessiki and Pieroni [29] was used to calculate the associated yield deformation. The displacement ductility index can be calculated as (Figure 19):(10)$$\mu_{\Delta }=\frac{\delta_{u}}{\delta_{y}}$$
where δ_u_ is the axial deformation at ultimate load, and δ_y_ is the axial deformation at yield load.

Similarly, the curvature ductility index can be calculated as:(11)$$\mu_{\Phi }=\frac{\Phi_{u}}{\Phi_{y}}$$
where Φ_u_ is the curvature at ultimate load, and Φ_y_ is the curvature at yield load.

The energy absorption capacity (W) of the beams was calculated as the area under the load versus the deformation curve for the mid-span location. The ductility factors and energy capacities for the beams are listed in Table 4.

Both ductility indices showed similar improvements. The deformation ductility index of the tube beam was improved by 20% compared to the control beam, while the curvature ductility index indicated 24% improvement. The energy absorption capacity of the tube beam was 1.34 times higher compared to the control beam.

## 7. Section Analysis

Section analysis of the beams was conducted using a simple fiber model that incorporated the concrete model described in Section 2.3. The compression zone of the tube beam was divided into unconfined and confined areas to approximate the way the beam was constructed. The top of the compression zone was modeled as an unconfined area, and below that for a depth equal to the grid tube diameter, the concrete was modeled as confined concrete, while the remaining area below that and closer to the neutral axis was also considered as unconfined (see Figure 20). For the control beam, the compression zone was treated as unconfined.

Some of the assumptions employed in this fiber model include perfect bond between concrete and the reinforcing bars (strain compatibility), and plain sections remain plain (linear strain distribution). The area below the neutral axis was considered cracked and was ignored in the force and moment calculations.

Each area of the compression zone was divided into thin strips, for which the compressive force was calculated as the product of the area of the strip and the average stress. The average stress was determined based on the stress–strain models developed in Section 2.3. The tensile force was calculated as the product of the area of steel reinforcement and the steel stress in each reinforcement layer by assuming an elastic, perfectly plastic stress–strain curve. The moment in the cross-section was determined by summing moments about the neutral axis.

The peak curvature was calculated as follows [27]:(12)$$\Phi =\frac{M}{E_{\mathrm{eff}}\cdot I_{\mathrm{eff}}}$$
where M is the moment in the section, E_eff_ is the effective concrete modules, as defined by Eurocode 2 (7.4.3 (6)), and I_eff_ is the effective moment of inertia of the section, as defined by Eurocode 2 (7.4.3 (3)) [2].

The model predictions on the peak load and curvature at peak load are shown in Table 5. The predictions for moment capacity were very close to the experimental values recorded, with a very small difference in the range of 2%. The predictions on the curvature were also relatively close, with differences ranging from 9% for the control beam and 12% for the tube beam.

In order to evaluate the effect of the CFRP grid on the curvature ductility index, a small parametric study was performed. The percentage of steel reinforcement was varied from 1% to 7%, and using section analysis, the moment and curvature were calculated. It should be noted here that the curvature was calculated as the ratio of the strain in the extreme concrete fiber in compression over the depth of the neutral axis. Then, curvature ductility indices were calculated in the same manner described at the beginning of Section 4. The results are presented in Table 6.

From the parametric study results, it seems that as the reinforcement ratio increases, the ductility index is reduced for both the control and grid beams. No significant change in the ductility index was observed beyond the 6% reinforcement ratio. The improvement in ductility index between the control and grid beams follows the same trend. There was a larger improvement for the lower reinforcement ratios compared to the higher reinforcement ratios. The tension-controlled beams had higher ductility compared to the compression-controlled beams, which was in line with design codes. It should be noted that the tested beams had a reinforcement ratio of approximately 6%. The model predicted a ductility index improvement of 34%, while the experimental ductility index was improved by approximately 25% for the tube beam compared to the control beam. This represents a difference of approximately 9% between the model prediction and the experimental data. This difference was considered acceptable.

## 8. Conclusions

In this study, a model predicting the confinement effect of a particular CFRP grid was presented along with the test results of compression-controlled concrete beams using CFRP grid tubes in the compression zone as concrete confinement reinforcement. Based on the findings presented, the following conclusions were drawn:The CFRP grid provides light confinement to concrete with the post-peak portion of the stress–strain curve having a descending branch similar to the results obtained by other researchers for lightly confined concrete.The confinement model developed predicted the behavior of concrete confined by CFRP grids with a high degree of accuracy.The results from the two beams indicate that the CFRP grid tubes provided confinement to the concrete. The use of a series of tubes rather than a large one was effective, and confinement was provided. This was evident from the small plateau that was formed when the load-displacement or moment-curvature curves of the tube beam were plotted.There was a nominal increase in the peak load of the tube beam of approximately 3% compared to the control beam. The load increase was considered minor.The ductility indices of the tube beam were 20–24% higher compared to the control beam. Similarly, the energy absorption capacity of the tube beam was improved by 35% compared to the control beam. This was in line with other compression-controlled specimens employing the same grid as confinement reinforcement for concrete tested by Michael [30]. The improvements observed were very similar.The model used to predict moments and curvatures for the beams was fairly accurate since it predicted the peak moment with high accuracy and the associated curvatures with relatively good accuracy. It also predicted the improvement in the ductility index for the beams.The parametric study indicated the obvious, that confinement of the compression zone improves ductility indices for the beams for which failure was initiated through steel yielding. The combination of the additional capacity for crushing strain due to confinement and steel yielding resulted in very significant improvements. The improvement reduces as the percentage of steel reinforcement increases. However, improvement was observed even with high percentages of steel reinforcement in compression-controlled members due to the confinement of concrete from the CFRP grid tubes.The CFRP grid used in this study had relatively low mechanical properties. It is anticipated that if grids with superior properties are used, the confinement effect will improve. If that is also combined with larger amounts of the CFRP material, significantly larger improvements can also be expected. Further investigation using such CFRP grids is needed since this type of application has not been investigated by many researchers.

## Figures and Tables

**Figure 1 materials-14-05163-f001:**
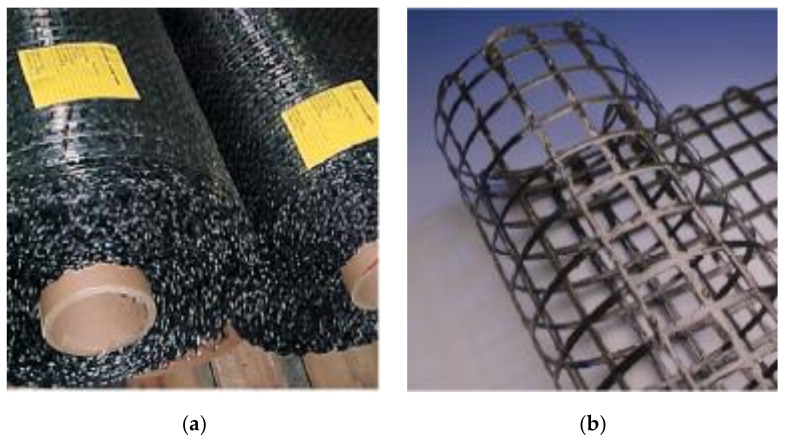
(**a**) CFRP grid rolls as received from the factory, and (**b**) CFRP grid formed into a tubular (coil) configuration (most effective configuration for concrete confinement) (photos courtesy of TechFab).

**Figure 2 materials-14-05163-f002:**
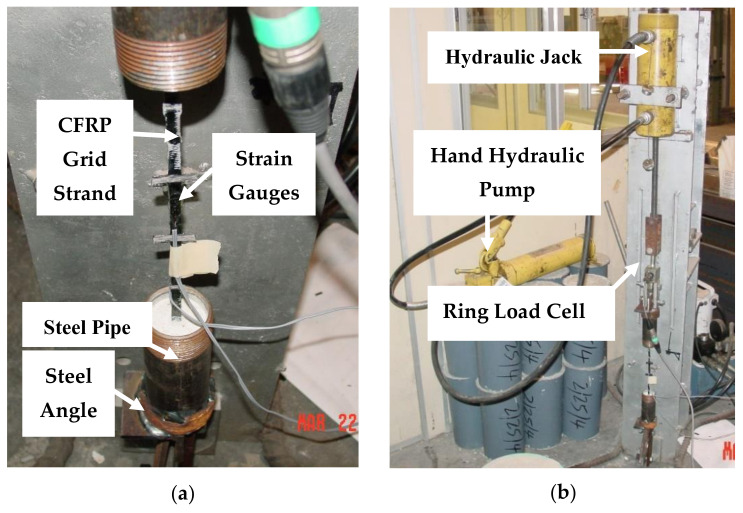
(**a**) CFRP grid stand specimen anchored using an expansive grout inside a steel pipe attached to the loading set-up using steel angles, and (**b**) test set-up for tensile testing, with a CFRP grid strand specimen attached to the loading frame using a hydraulic jack to apply force and a ring load cell to measure the total force applied [3].

**Figure 3 materials-14-05163-f003:**
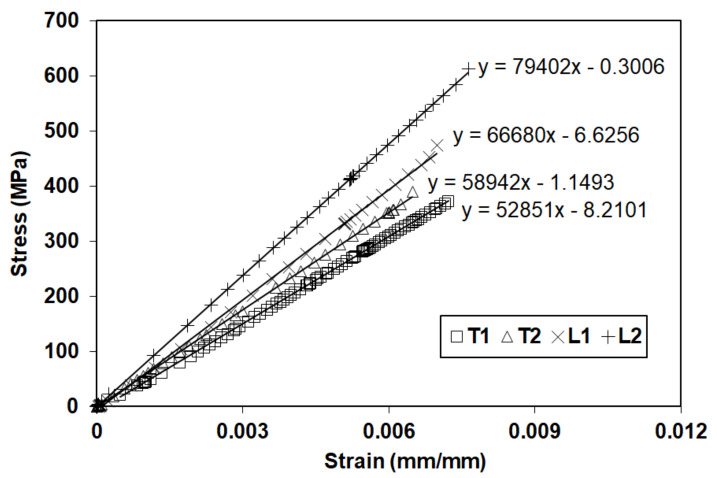
Stress-strain curves of CFRP grid stands obtained from axial tensile tests with linear regression lines and the equations of these lines used to determine modulus values (slope of each line) [3].

**Figure 4 materials-14-05163-f004:**
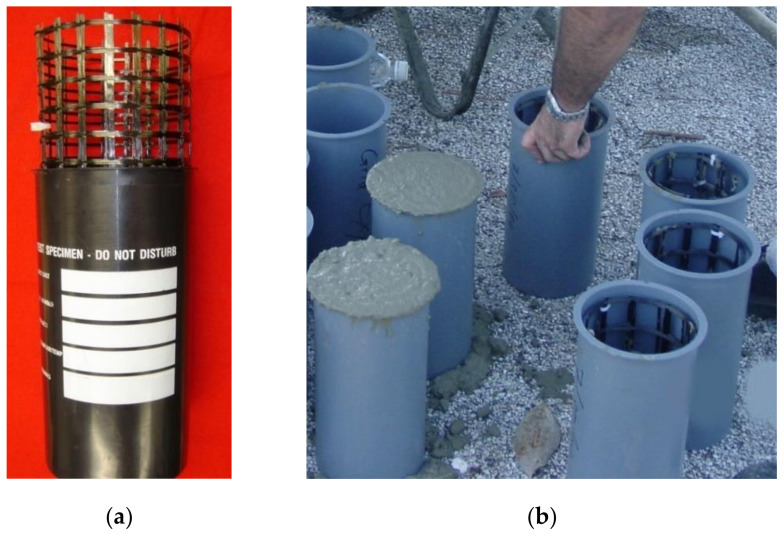
CFRP grid cylinder manufacturing: (**a**) CFRP grid tube (coil) inside a cylindrical mold prior to casting and (**b**) CFRP grid tubes inside the cylindrical molds during concrete casting [3].

**Figure 5 materials-14-05163-f005:**
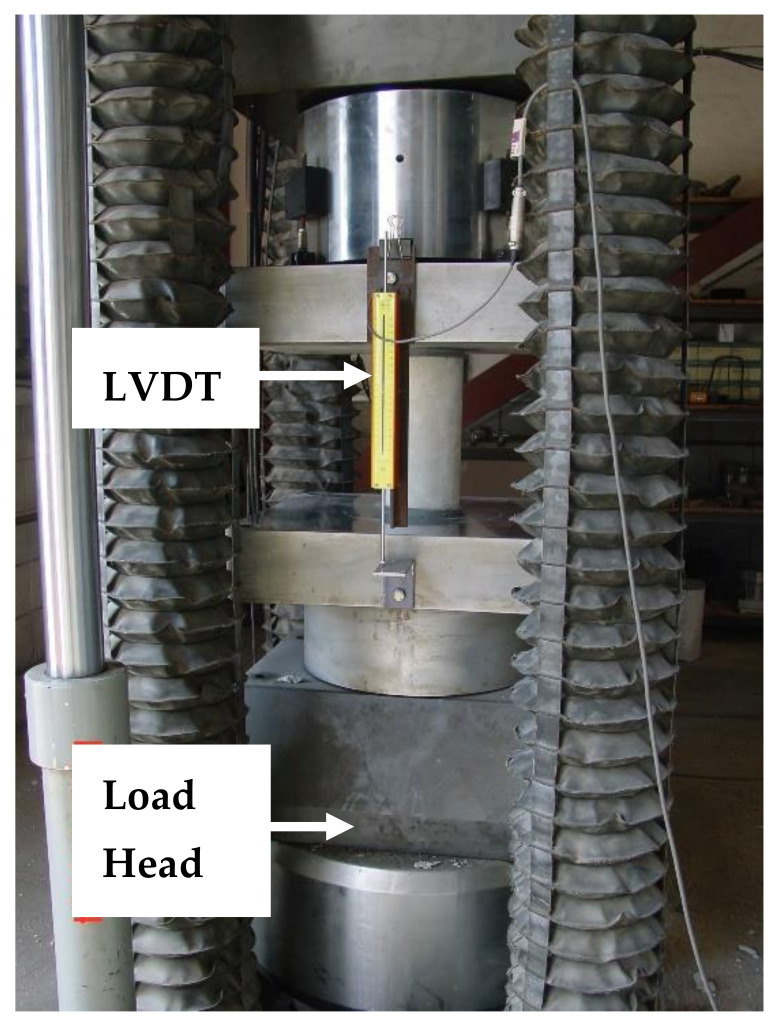
CFRP grid cylinder inside the loading frame with measuring instruments (load cell and displacement transducers (LVDTs)) [3].

**Figure 6 materials-14-05163-f006:**
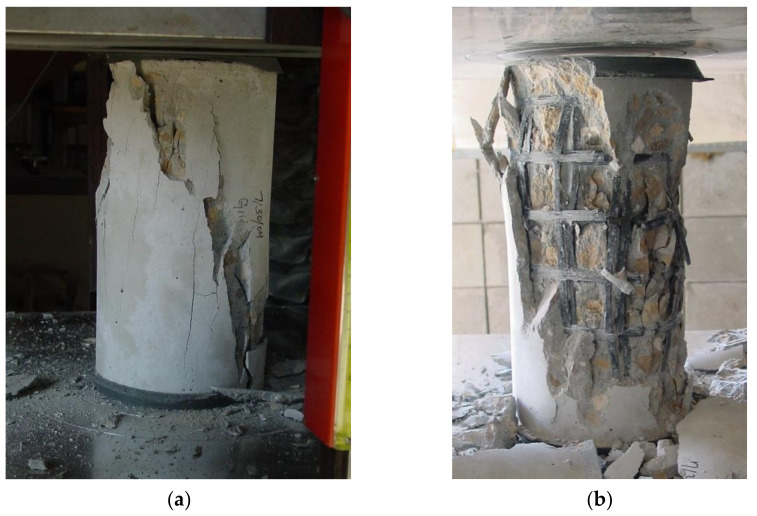
Cylindrical specimens after testing: (**a**) typical failure for control specimens, and (**b**) typical failure of grid specimens caused by rupture of CFRP grid strands [3].

**Figure 7 materials-14-05163-f007:**
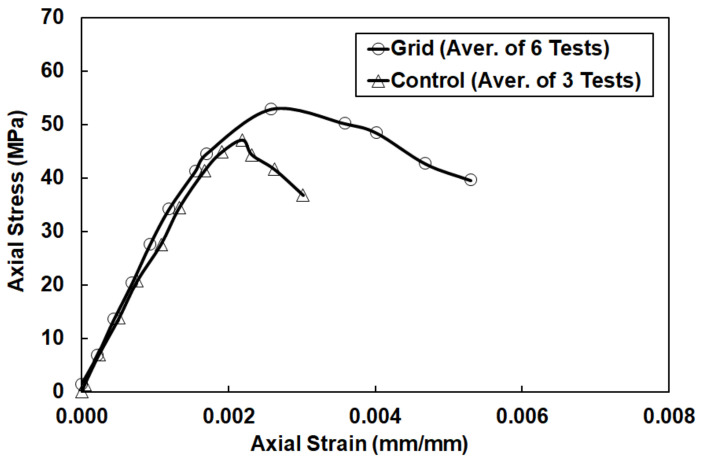
Average experimental stress–strain curves for control and grid specimens.

**Figure 8 materials-14-05163-f008:**
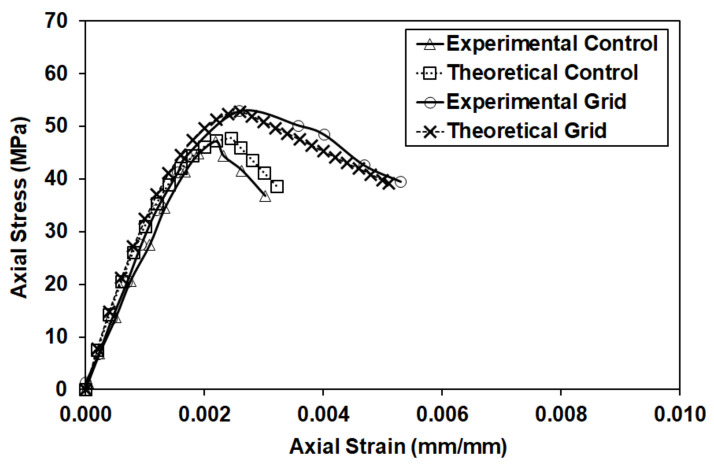
Average experimental and theoretical (confinement model) stress–strain curves for control and grid specimens.

**Figure 9 materials-14-05163-f009:**
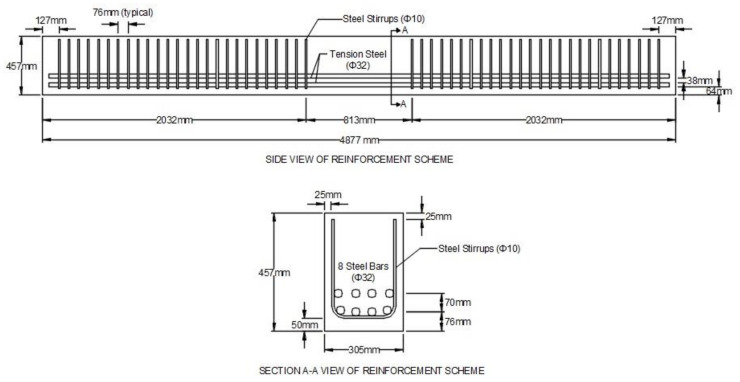
Side (top part of figure) and cross-sectional (bottom part of figure) views of reinforcement scheme for control beam showing longitudinal tension steel (Φ32 bars) and steel vertical U-shaped stirrups (Φ10 bars).

**Figure 10 materials-14-05163-f010:**
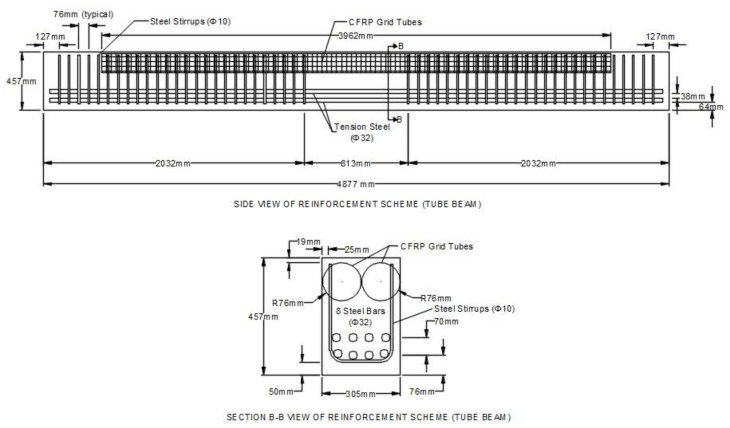
Side (top part of figure) and cross-sectional (bottom part of figure) views of reinforcement scheme for tube beam showing longitudinal tension steel (Φ32 bars), steel vertical U-shaped stirrups (Φ10 bars), and 2 CFRP grid tubes (coils) at the top of the beam used to confine concrete in the compression zone.

**Figure 11 materials-14-05163-f011:**
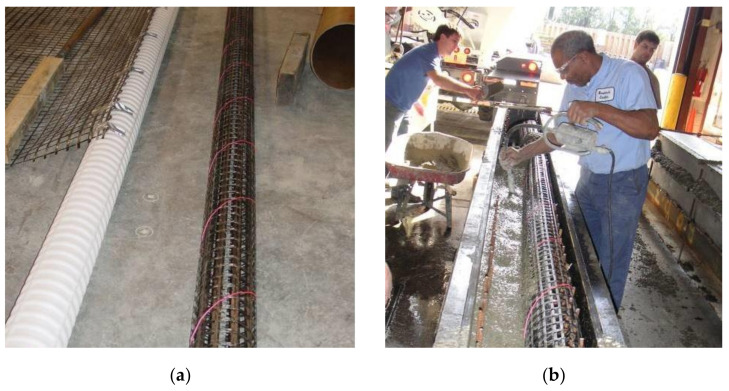
(**a**) Tube (coil) of CFRP grid formed using a plastic pipe held in place by a string, and (**b**) concrete casting of the tube beam (initially only 1 CFRP tube (coil) was placed to facilitate concrete pouring and leave space for the use of a vibrator).

**Figure 12 materials-14-05163-f012:**
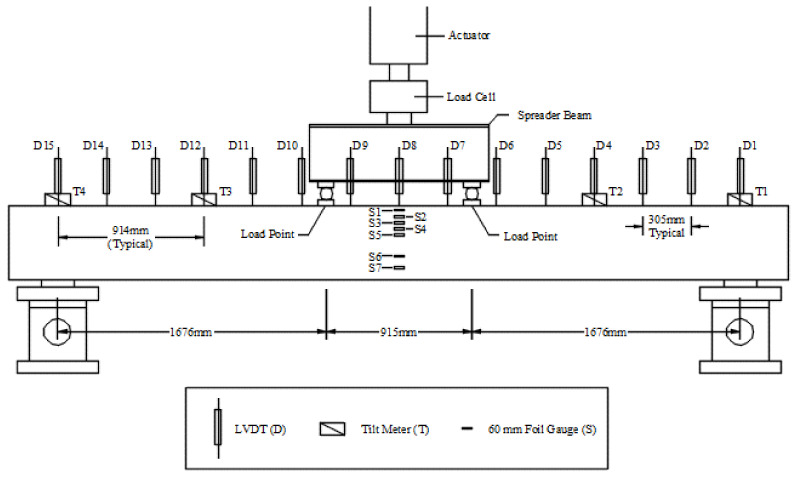
Schematic of the test set-up showing location of instruments used to measure load, displacement, and strains, as well as location of loading points with respect to supports.

**Figure 13 materials-14-05163-f013:**
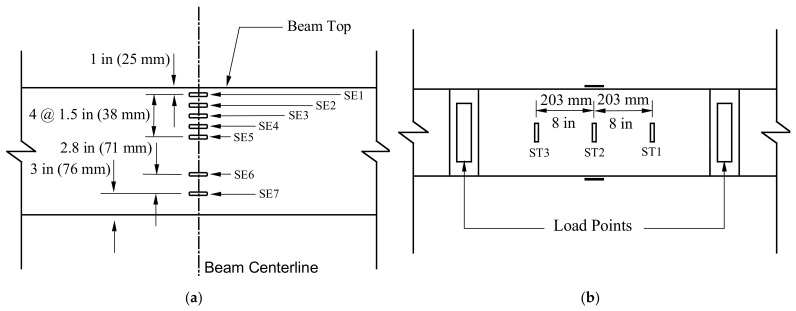
Strain gage arrangement at beam mid-span: (**a**) east side of the beam, and (**b**) top of the beam.

**Figure 14 materials-14-05163-f014:**
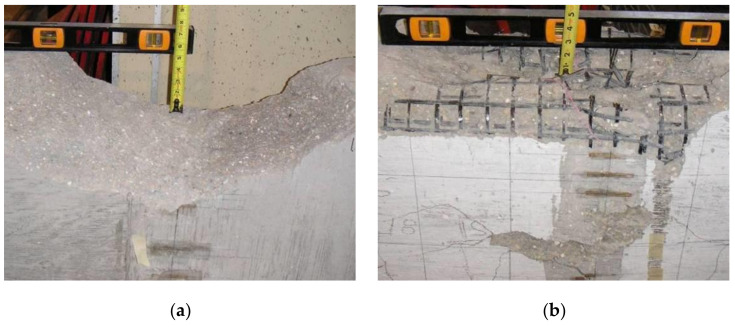
Compression zone damage after testing (**a**) control beam, and (**b**) tube beam.

**Figure 15 materials-14-05163-f015:**
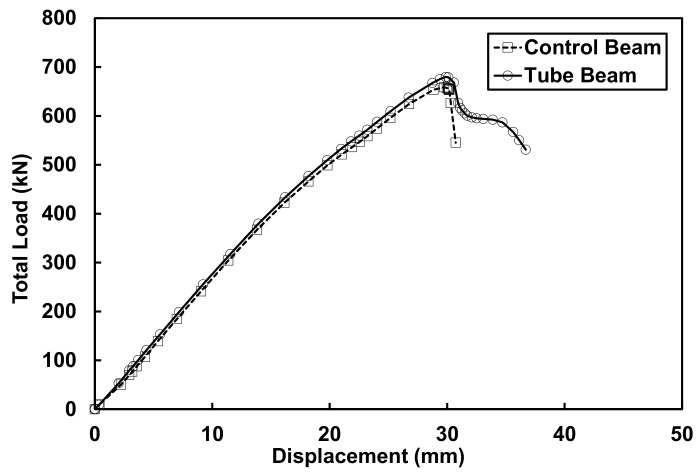
Experimental load-displacement curves for both control and tube beams.

**Figure 16 materials-14-05163-f016:**
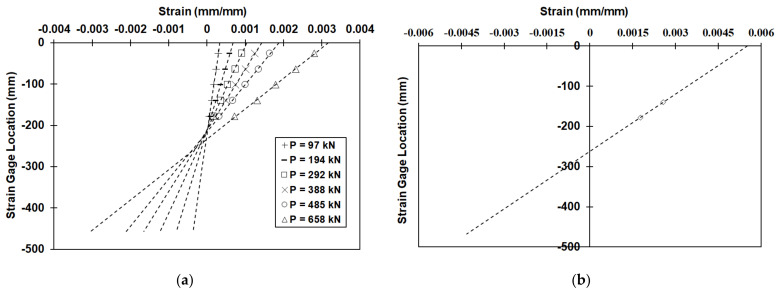
(**a**) Strain distribution profiles across the depth for control beam at various loads, P, based on measurements from strain gages, and (**b**) strain distribution profile for tube beam across the depth at load causing crashing of confined concrete (post-peak load 567 kN), based on available strain data.

**Figure 17 materials-14-05163-f017:**
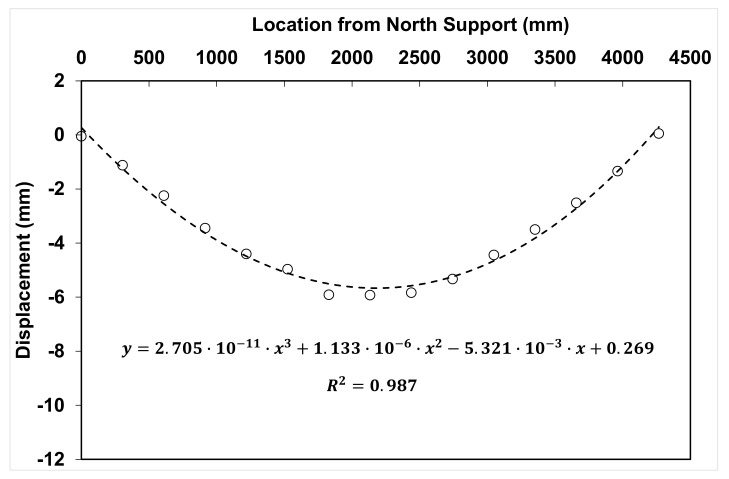
Deflection profile for control beam based on LVDT recorded measurements (circular marks) and deflection equation based on a third order polynomial trend line fitted to experimental data for a load P = 194 kN.

**Figure 18 materials-14-05163-f018:**
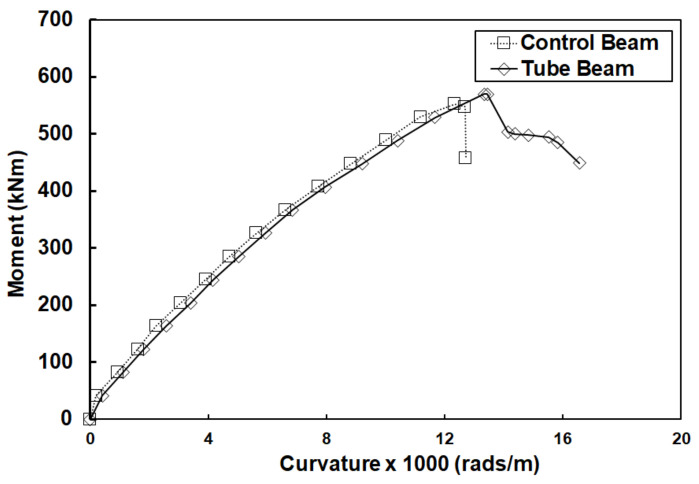
Experimental moment-curvature curves at mid-span for both control and tube beams. Values of curvature were calculated based on the second derivative of the deflection equations (as shown in Figure 17) at various loads.

**Figure 19 materials-14-05163-f019:**
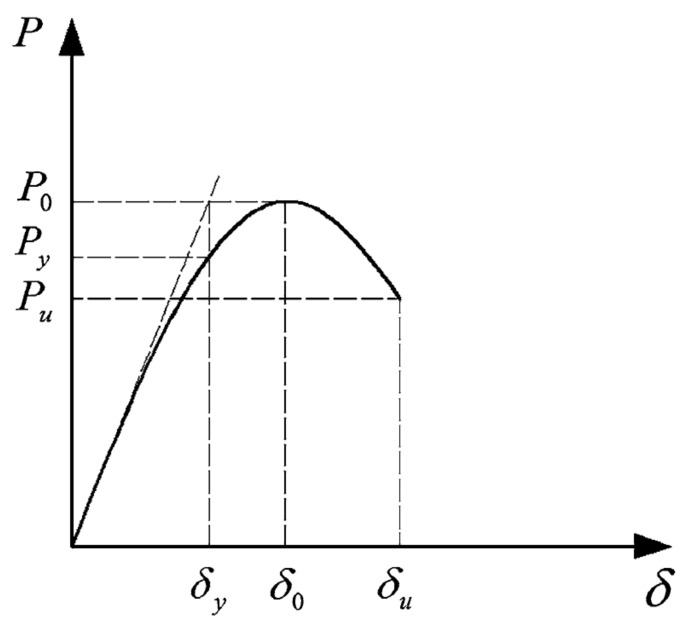
Ductility index component definition used to calculate yield and ultimate deflections and curvatures for the determination of ductility indices [4].

**Figure 20 materials-14-05163-f020:**
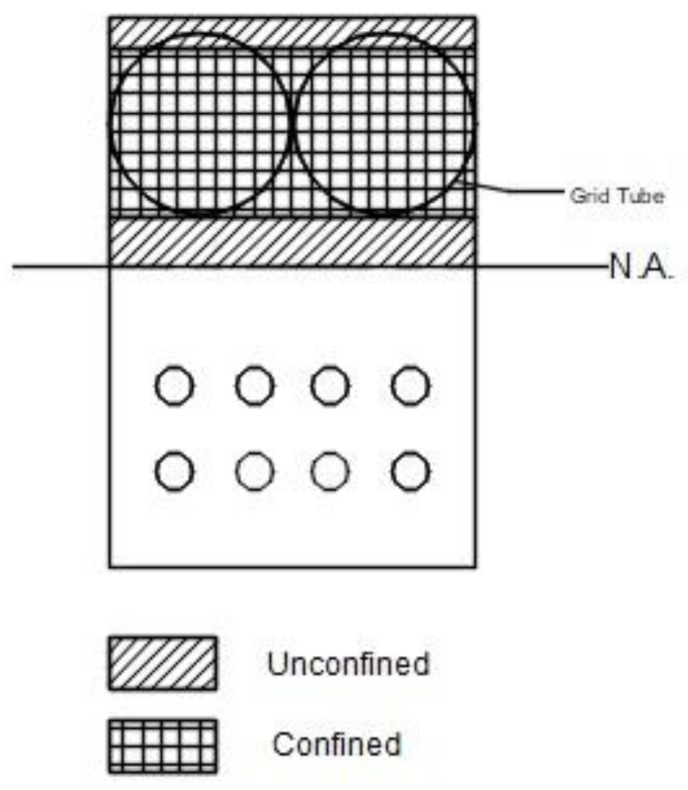
Tube beam cross-section used in fiber model, indicating confined and unconfined concrete regions.

**Table 1 materials-14-05163-t001:** Experimental results for CFRP grid specimens [3].

Specimen	Direction	Area (mm^2^)	Peak Load (kN)	Strength (MPa)	Modulus (GPa)
T1	Transverse	7.91	3.74	472.5	52.9
T2	Transverse	5.86	5.10	870.2	58.9
L1	Longitudinal	6.08	4.11	675.5	66.7
L2	Longitudinal	5.18	3.96	763.8	79.4
Average	N/A	6.26	4.23	695.5	64.5

**Table 2 materials-14-05163-t002:** Experimental results for peak load, strain at peak load, and peak stress for control and CFRP grid specimens [3].

Type	Grid Diam. (mm)	Peak Load (kN)	Strain at Peak Load (mm/mm)	Conc. Core Area (mm^2^)	Peak Stress (MPa)	Avg. Peak Stress (MPa)	COV (%)
Control 1	N/A	854.8	0.00251	18,241	46.9	47.8	2
Control 2	N/A	853.2	0.00232	18,241	48.8		
Control 3	N/A	870.6	0.00236	18,241	47.7		
Grid 1	133.4	820.8	0.00281	13,966	58.8	52.9	8.6
Grid 2	133.4	650.7	0.00200	13,966	46.6		
Grid 3	133.4	792.6	0.00256	13,966	56.8		
Grid 4	139.7	743.5	0.00279	15,328	49.6		
Grid 5	139.7	775.9	0.00256	15,328	51.8		
Grid 6	139.7	803.7	0.00279	15,328	53.7		

**Table 3 materials-14-05163-t003:** Experimental results for unconfined concrete cylinder specimens used to evaluate the concrete strength of beams.

Cylinder No.	Peak Load (kN)	Peak Stress (MPa)	Avg. Peak Stress (MPa)	COV (%)
1	765.8	42.0	41.8	2.3
2	782.4	42.9		
3	732.6	40.2		
4	770.3	42.2		
5	745.2	40.9		
6	769.9	42.2		
7	748.6	41.0		
8	778.0	42.7		

**Table 4 materials-14-05163-t004:** Yield and ultimate deflections and curvatures determined from experimental curves, ductility indices, and energy capacities for control and tube beams.

Specimen	δ_y_ (mm)	δ_u_ (mm)	Φ_y_ (rads/m)	Φ_u_ (rads/m)	μ_Δ_	μ_Φ_	W (kNmm)
Control Beam	25.2	30.7	9.5 × 10^−3^	12.8 × 10^−3^	1.22	1.35	11,307
Tube Beam	25.1	36.7	9.2 × 10^−3^	15.5 × 10^−3^	1.46	1.68	15,160

**Table 5 materials-14-05163-t005:** Model predictions and experimental values for peak load and curvature at peak load.

Specimen	M_exp_ (kNm)	M_model_ (kNm)	Φ_exp_ (rads/m)	Φ_model_ (rads/m)	M_model_/M_exp_	Φ_model_/Φ_exp_
Control Beam	552	539.5	12.4 × 10^−3^	11.3 × 10^−3^	0.98	0.91
Tube Beam	569.7	561.4	13.3 × 10^−3^	11.7 × 10^−3^	0.98	0.88

**Table 6 materials-14-05163-t006:** Parametric study results for various steel reinforcement ratios and associated curvature ductility indices for control and tube beam cases.

Steel Reinf. Ratio (%)	Φ_yc_ (rads/m)	Φ_uc_ (rads/m)	Φ_yt_ (rads/m)	Φ_ut_ (rads/m)	$\mu_{\Phi c}=\frac{\Phi_{yc}}{\Phi_{uc}}$	$\mu_{\Phi t}=\frac{\Phi_{yg}}{\Phi_{ut}}$	$\frac{\mu_{\Phi t}}{\mu_{\Phi c}}$
1	8.59 × 10^−3^	60.21 × 10^−3^	9.23 × 10^−3^	164.19 × 10^−3^	7.01	17.79	2.54
2	10.12 × 10^−3^	31.15 × 10^−3^	10.29 × 10^−3^	66.08 × 10^−3^	3.08	6.42	2.08
3	10.63 × 10^−3^	22.05 × 10^−3^	10.74 × 10^−3^	41.91 × 10^−3^	2.07	3.90	1.89
4	11.02 × 10^−3^	17.48 × 10^−3^	11.77 × 10^−3^	30.54 × 10^−3^	1.59	2.59	1.63
5	10.67 × 10^−3^	15.74 × 10^−3^	11.28 × 10^−3^	23.29 × 10^−3^	1.48	2.07	1.40
6	10.23 × 10^−3^	14.54 × 10^−3^	11.51 × 10^−3^	21.85 × 10^−3^	1.42	1.91	1.34
7	9.81 × 10^−3.^	13.99 × 10^−3^	10.57× 10^−3^	20.21 × 10^−3^	1.42	1.91	1.34

Φ_yc_ is the yield curvature for the control beam, Φ_uc_ is the ultimate curvature for the control beam, Φ_yt_ is the yield curvature for the tube beam, Φ_ut_ is the ultimate curvature for the tube beam, $\mu_{\Phi c}$ is the curvature ductility index for the control beam, and $\mu_{\Phi t}$ is the curvature ductility index for the tube beam.

## Data Availability

Data sharing is not applicable to this article.

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
