# Peer review of "Experimental Ductility of Compression-Controlled Flexural Members Using CFRP Grid to Confine Concrete"

_materials, 2021, doi:10.3390/ma14185163_

Round 1

Reviewer 1 Report

This work proposes an approach towards improving the flexural behavior of over-reinforced members via concrete confinement with the help of CFRP grid tubes in the compression zone. The idea is a very thorough using experimentation. Ductility of the beams seem to increase up to 30%, when compared, to control work. I have the following questions for the authors:

1) How were the displacements and ductility indices  calculated using the experimental work?

2)  What would happen, if the strain compatibility assumption in the fiber model is changed?

Reviewer 2 Report

The paper introduced a new approach that using CFRP grid tubes in the compression zone to improve the flexural behavior of the concrete. According to the Experimental results, the ductility of the tube beam increased by 20%-30%, the improvement is indeed significant. The author has done a meaningful work, which is worthy of recognition.

1. It is recommended to shorten some long sentences in the text.

2. In Fig. 8, the author stated that “the modified Hognestad curves matched well with the experimental curves”. But I think the experimental and theoretical control curves don’t match well, the slopes at the beginning of two curves are quite different, why?

3. The serial number of the subtitle in the article is not correct.

4. The materials and material testing part of the article cited a lot of data of the paper [3], is it necessary to use so much data in part 2, could it be shortened?

Reviewer 3 Report

While the use of CFRP grids is not novel, this local application for flexure does not seem to have been well investigated. The paper is well written and all procedures/experiments and models carefully explained.

The biggest weakness in presentation is the figures which in some cases are of poor quality (fonts too small, unreadable legends/axes -  Fig 17),  not as clearly labeled as they could be (fig 9 and 10 - label materials on the figure, is the stirrup CFRP as well? in the caption Side(Top) is confusing since side and top are different things in this work) or not as useful as they could be (Fig1(a), is this a coil of the grid?)

Other suggestions

  • define balanced, over and under-reinforced
  • CFRP on first use (rather than 2nd)
  • Line 175 - concentric tubular configuration does this mean that there were multiple concentric rings of the grid inserted in the beam? later only one diameter (for each set of beams) is mentioned - not clear 
  • a figure of the grid in the beam/cylinder - maybe top view - would be helpful - make Fig 4 a close up of the cylinders? otherwise it's not a particularly useful figure. (Fig 11 is useful)
  • All captions could be longer and provide more detail - even if it repeats what is in the test.
